# The Challenges of Inferring Organic Function from Structure and Its Emulation in Biomechanics and Biomimetics

**DOI:** 10.3390/biomimetics6010021

**Published:** 2021-03-18

**Authors:** Karl J. Niklas, Ian D. Walker

**Affiliations:** 1School of Integrative Plant Science, Cornell University, Ithaca, NY 14853, USA; 2Department of Electrical and Computer Engineering, Clemson University, Clemson, SC 29634, USA; iwalker@g.clemson.edu

**Keywords:** adaptation, engineering theory, evolution, form-function, modelling, plants

## Abstract

The discipline called biomimetics attempts to create synthetic systems that model the behavior and functions of biological systems. At a very basic level, this approach incorporates a philosophy grounded in modeling either the behavior or properties of organic systems based on inferences of structure–function relationships. This approach has achieved extraordinary scientific accomplishments, both in fabricating new materials and structures. However, it is also prone to misstep because (1) many organic structures are multifunctional that have reconciled conflicting individual functional demands (rather than maximize the performance of any one task) over evolutionary time, and (2) some structures are ancillary or entirely superfluous to the functions their associated systems perform. The important point is that we must typically infer function from structure, and that is not always easy to do even when behavioral characteristics are available (e.g., the delivery of venom by the fangs of a snake, or cytoplasmic toxins by the leaf hairs of the stinging nettle). Here, we discuss both of these potential pitfalls by comparing and contrasting how engineered and organic systems are operationally analyzed. We also address the challenges that emerge when an organic system is modeled and suggest a few methods to evaluate the validity of models in general.

## 1. Introduction

Attempts to model or emulate organic structure–function relationships often rely on the fact that the physiological and structural requirements for growth, survival, and reproductive success are remarkably similar for the majority of extant as well as extinct organisms regardless of their phyletic affiliation, even at the molecular level [1,2,3,4,5]. In addition, most of these requirements can be quantified by means of comparatively simple mathematical relationships drawn directly from the physical and engineering sciences [6,7,8,9,10,11]. Owing in part to the advent and rapid expansion of computational and fabrication technologies, the number of structure–function models has burgeoned in the last few decades to encompass every level of biological organization ranging from molecular self-assembly (e.g., in vitro spindle apparatus self-assembly) to ecological and evolutionary dynamics (e.g., diversification patterns resulting from differential extinction–origination models) (e.g., [2,10,12,13]).

The interest in modeling organic structure–function relationships has expanded perhaps most aggressively in the field of biomechanics and subsequently into biomimetics—an interdisciplinary discipline in which researchers apply the principles of physics, engineering, chemistry, mathematics, and biology to create synthetic inorganic systems or materials that mimic the functions of biological systems, materials, or processes [3,4,14,15]. This approach has resulted in significant progress in both its theoretical and practical applications. This success emerges in part because biological systems have passed the gauntlet of evolutionary time and successive episodes of natural selection, thereby winnowing out failures. In this sense, almost every organic system can be viewed as the result of a series of evolutionary ‘experiments’ whose continued success reflects the consequences of intense scrutiny [9]. However, extracting insights from these ‘experiments’ and mimicking the attributes of organic systems presents a number of challenges not least of which is the assumption that structure–function relationships can be inferred correctly and unambiguously.

In addition, by their very nature (as well as their specific epithets), the fields of *bio*mechanics and *bio*mimetics rely heavily on the principles and theoretical insights gained from the application of engineering and physics to analyze biological system in terms that differ in many important ways from those that emerge directly from the study of biological systems. Indeed, as the level of biological enquiry increases from the molecular to the organismic and from the organismic to the level of evolutionary processes, the ‘gap’ between the physical and biological sciences increases non-linearly and not without substantial practical and theoretical consequences and challenges.

One of the goals of this paper is to compare and contrast the philosophical differences emerging from the physical sciences (specifically engineering and physics) and the ‘synthetic’ fields of enquiry called biomechanics and biomimetics. The applicability of the fundamental principles of the physical sciences to understand biology is beyond question. No organism can obviate these principles. However, it must be acknowledged that the manner in which these principles describe or predict the behavior of biological systems differs from how they apply to inorganic systems.

Consider for example the concept of entropy. Although it has two very formal definitions (one in the context of thermodynamics and another in terms of statistical mechanics), entropy can be viewed as a measure of a system’s disorder, or randomness. In simple terms, the basic principle associated with entropy (the Second Law of Thermodynamics) states that in the absence of any externally applied source of energy, a system will naturally progress from an ordered to a disordered state. Thus, for any process to occur spontaneously, it is necessary that the entropy in the system in which the process occurs decrease. Living things appear to be negentropic—that is, they appear to become more ordered and less random. This seems to be the case when we look at a living cell or a multicellular organism as it grows in size and continues to function normally. Yet, when we consider that every living organism is an open system—that it exchanges matter and energy with its immediate environment—we see that this negentropic perception is incorrect. All living organisms increase the entropy of their environment in two ways—they convert larger organic molecules (e.g., proteins, carbohydrates, and fats) into smaller inorganic molecules (e.g., CO_2_, water, NH_3_), and they produce heat. The decrease in entropy associated with the growth of an organism is greatly outweighed by the increase in entropy in the organism’s environment. A simple example of this is the so called “10% rule”, which governs the flow of biomass between two adjacent trophic levels in an ecosystem. Each pound of living “stuff” that is made in the next higher trophic level requires the metabolic conversion of 10 pounds of living matter (into CO_2_, water, and heat) from the lower trophic level. Thus, only 10% of the material produced in the lower trophic level appears as biomass in the higher level.

In this paper, some emphasis will be placed on the implications emerging from the use of models in biomechanics and biomimetics. In this context, a model is defined as any physical or conceptual representation of a naturally occurring system or process intended to provide mechanistic insights into the operation of the phenomenon under investigation. As will be seen, this definition distinguishes a model from a prototype or pattern-architype (wherein the functionality of the prototype or pattern-architype is irrelevant to how the prototype or pattern-architype is used). The benefits of modeling physical or biological systems are widely acknowledged across every scientific discipline. This general applicability results from the fact that many systems are unavailable for direct observation or experimentation because they are too big, small, expensive, rare, or fragile to observe or manipulate directly. Modeling such systems to analyze their behavior is a convenient approach to coping with these limitations. Likewise, some systems that are subject to direct observation may be so complex or imperfectly known that modeling provides the only tractable method to explore the details of their operation. Here, models can be used to generate hypotheses that can be empirically examined subsequently. Finally, models are an important part of our pedagogic armamentarium. They provide a method to communicate and summarize complex ideas or operations easily yet accurately. However, it must be acknowledged that as an abstraction (or perhaps more accurately a redaction) of reality, models can obtain fallacious results. Perhaps worse, a model can give the correct answers but for the wrong reasons, e.g., the astrolabe of Claudius Ptolemy (~100–170 CE) yielded correct predictions about lunar phenomena, but nevertheless assumed the Earth was the center of the universe. This potential pitfall is important when the objective of a model is to elucidate mechanistic explanations for a given phenomenon. It must be freely admitted that a mechanistically incorrect model inspiring an engineer or designer to create a novel and ultimately successful solution to a problem is a useful and valuable model. The “trick”, however, is to know whether the model is mechanistically valid or invalid.

In the following sections, we discuss the types of models that are typically used in science, compare and contrast the approaches taken in the physical and biological sciences, and suggest criteria that can be used to assess engineering and biological models in general.

## 2. An Important Caveat about Behavior

It is important to note that the bulk of what follows focuses on plants rather than animals. Acknowledging this focus is essential because the discussion to be presented largely ignores the important role of “behavior” when inferring function from structure. Animals typically manifest behavior that is immediately obvious (and typically easily translated into human terms), whereas plant “behavior” is generally not immediately obvious because it operates on extended time-scales (i.e., over the course of hours, days, or even years) and primarily at the cellular rather than supracellular level. Consider for example the hypodermic functionality of a snake’s fang and the unicellular epidermal hairs (trichomes) on the leaves of the common stinging nettle *Urtica dioica*. Both of these structures are capable of injecting a toxic substance subcutaneously by means of exerting pressure on a containment vesicle (i.e., the venom gland and the cell base of the trichome). The snake delivery system accomplishes this by means of provoked neuromuscular contraction (an act of volition) of a complex multicellular venom gland, whereas the nettle injects its cellular contents passively when its acicular tip is fractured and broken off. Arguably, at many levels, the fang and trichome have very little structural resemblance to a medical hypodermic needle. However, the pressure and fluid flow behaviors of all three are similar upon reflection, but only after observing how all three behave dynamically. In the case of the trichome, this kind of observation would require careful microscopic manipulations (which, in turn, necessitate premeditation on the part of the researcher and some degree of fore-knowledge).

Far more important in the context of this paper is a dictum that emerges from what can be called the “no function in structure” principle, which essentially argues that “function” resides in the “behavior” of a system rather than in the “structure” of a system. According to this principle, structure *implements* function, just as a medical hypodermic needle *injects* a liquid or a fang injects venom, the *behavior* of the needle or the fang is in fact the *function* of the needle and the fang. This principle argues that function can only be achieved by the behavior of a system, which can be redacted by examining how parameters, properties, or some other variables of interest change and not on the basis of examining structure. In the context of dynamic systems, the “no function in structure” principle is relevant to both engineering and biology. However, it misses an important point, viz., evolution and natural selection operate on the manifest, integrated phenotype and the phenotype includes structure as well as behavior. In addition, there are many aspects of plant biology in which structure dictates behavior and in which structure has no behavior (e.g., the material properties, structure, distribution, and conformations of the different cell-types in wood dictate the behavior of wood, and the functionalities of a thorn do not involve any manifest behavior). We shall return to this issue but raise it here to keep the “no function in structure” principle in sight.

## 3. Types of Models

What constitutes a model and how can a model be tested? This question is not easily answered because models come in many forms, are put to different uses, and must survive the challenge of different kinds of tests to gain credibility. Although there are many models, there are only three general types: iconic models, analogue models, and mathematical models [12]. Iconic models are physical conceptions of reality (e.g., scale models). Analogue models are schematic representations of dynamic processes or operations (e.g., photosynthesis rendered in chemical notation). Mathematical models are the most abstract of the three general types because they express reality in terms of computational operations (e.g., the Nernst equation). Importantly, each type of model can be used as a descriptive, behavioral, or decision-making tool. A descriptive model represents the operational relationship, order, or sequencing among the components of a real system (e.g., a reconstruction of a plant used to show organographic or anatomical relationships). Behavioral models are used to predict the response of a real system to perturbation (e.g., a scale model of plant placed in a wind tunnel to estimate wind-induced drag forces and stresses). Decision-making models are used to identify which among available alternative responses of a system is the most likely or favorable according to a priori criteria (e.g., cladistic algorithms and statistical inference models). Complex models can be constructed using two or more types of model in combination as descriptive, behavioral, and decision-making tools.

Regardless of its type or intended use, however, every model is a conceived image of reality and every modeler, therefore, must deal with the ongoing tension between the ideal (conceived image) and the real (observed data) (see [12]). This does not imply that all useful models are complex. A model’s complexity depends as much on the objective of modeling as it does on the structure, object, operation, or process being modeled. When put to some simple use, a model can successfully mimic a structure or process with a minimum number of assumptions or stipulations. For example, to evaluate convective cooling, a cow can be modeled as a cylinder (provided its surface area with respect to volume is scaled properly). Indeed, some very useful models bear little resemblance to the actual appearance of the object or system being represented. But it is always true that a model is only as useful as the extent to which its behavior or operation faithfully accords with that of reality. A cylindrical cow is a useful model in terms of understanding the physics of heat dissipation, but, in terms of locomotion, a cylinder rolls whereas a cow walks.

## 4. Plant Versus Animal Models

Modeling the relationship between animal structure–function relationships has a long and distinguished history (e.g., [16,17,18,19,20,21]). In comparison, modeling has only recently been used to explore the quantitative relationships between plant structure and function. Perhaps not coincidentally, efforts in biomimetics founded on plant structure–function relationships [22,23] have tended to lag behind those in animals [17,18,19,20,21,24]. While this is likely due in part by a desire to emulate various types of mobility—significantly hindered by the sessile nature of plants—we speculate that the relatively recent emergence of structure–function plant models has contributed to this lag. Nonetheless, within the last few decades, virtually every level of biological organization has been explored for plants, ranging from the level of molecular self-assembly to community structure, ecological interactions, and evolutionary dynamics.

Although the reasons for this burgeoning interest in modeling and mimicking plant structure–function relationships are obvious, the reasons that plant models are latecomers are not obvious, especially since modeling how organic structure and function interrelate is arguably easier for plants than for animals. Unlike animals, the vast majority of plant species perform the same tasks to grow, survive, and reproduce—plants use sunlight to manufacture their living substance, exchange atmospheric gases with their external environment, absorb and transport water and minerals, cope with the mechanical stresses induced by externally applied mechanical forces, and they capture or disperse spores, seeds, fruits, or other similar structures to reproduce or colonize new sites. Thus, unlike the case for animals, the metabolic operations and requirements of plants, both past and present, are nearly identical in their broad outline. Likewise, most plants are sedentary and all lack neurological and muscular systems, and thus they grow, reproduce, and die in much the same location in which they began their existence.

Therefore, plants can be modeled more as “structures” than as “mechanisms” (e.g., [5]), although plants manifest dynamic mechanical behaviors (e.g., [14]). For these reasons, the botanist can turn directly to the techniques and concepts formulated by the physical and engineering sciences to model virtually any plant function requiring energy or mass transport (e.g., photosynthesis, respiration, water and mineral absorption) or involving solid and fluid mechanics (e.g., heat dissipation, mechanical stability, passive or active dispersal of spores or propagules, and the bulk transport or movement of water or air). Analytical geometry can be used to evaluate the capacity of a plant to intercept sunlight; equations such as Fick’s law can be used to model the passive diffusion of carbon dioxide, oxygen, or any other substance through its tissues; the transport of water through plant vascular systems can be evaluated by means of the Hagen–Poiseuille equation; and the long-distance passive dispersal of reproductive structures by wind or water can be treated with the aid of Stokes’ law or comparatively simple physical or mathematical models. On a macroscopic scale, the ability of any plant to cope with the effects of gravity or drag induced bending moments and stresses can be evaluated with the aid of standard engineering techniques and concepts, whereas ecological interactions among conspecifics or among members of different plant species can be modeled often on the basis of simple allometric “rules” that appear to hold true through across the full spectrum of plant size and all clades.

The botanist is also at advantage because the spectrum of plant body-plans is narrower than that of animals [25]. Whereas the body-plans of the algae are extremely diverse (e.g., unicellular, colonial, filamentous, thalloid, and treelike), the land plant species share the same basic body-plan predicated on one or more cylindrical organs (stems or stem-like analogues) bearing flattened organs (leaves or leaf-like analogues) anchored to a substrate by a variety of means (e.g., rhizoids and multicellular scales) of which roots are the most familiar. Regardless of their apparent complexity, all plant body-plans can be modeled using one or more simple geometric forms (e.g., terete cylinders and oblate or prolate spheroids) whose shape and size can be adjusted independently or scaled with respect to each other.

Modeling any form–functional relationship, however, has its pitfalls. Whereas the fundamental tenet of modeling is that the first principles of physics, chemistry, engineering, and mathematics cannot be violated, it is also true that organisms typically obviate (or at least obscure) the effects of some of these well-known principles by virtue of growth, reproduction, or unique adaptations, aspects of which are sometimes poorly preserved or entirely obscured in the fossil record. For example, Fick’s law for passive diffusion always holds true, just as Euler’s buckling formula accurately predicts how tall a tree can grow before it elastically deflects under its own weight. But Fick’s law is irrelevant for species that actively transport carbon dioxide in the form of bicarbonates (as do some aquatic plants), whereas Euler’s buckling formula gives different estimates of the tree height for woods differing in their lignin content and thus stiffness (and assumes that mechanical constraints rather than hydraulic constraints limit tree height). Even the most credulous may reasonably doubt that organisms have properties that make the direct application of closed-form engineering solutions to understand their biology highly problematic.

## 5. Comparing Standard Practices (and a Case Study)

Physical principles can be used to interpret and understand biological phenomena, but it must be acknowledged that the standard practices used in the physical and biological sciences differ in many important ways (Table 1). Consider how an engineer approaches a challenge. In most cases, the working environment and design specifications for a machine, structure, or material are specified a priori. There are exceptions naturally. The design and construction of machines for deep-space or other-worldly exploration require considerably more thought. However, in general, there is no second guessing under most situations. In addition, the structure and the materials used to construct the required artifact can be altered during the design and implementation process. There is no historical constraint on design or construction practices. Finally, it is generally true (albeit not invariably so), that an engineered artifact has one or only a comparatively few functional obligations—a toaster, hair drier, phonograph, or comb has but one primary function. It must be recognized that a toaster or hair drier can be used to heat a room, a phonograph can be used to hypnotize an unwilling musical student, and a comb can be used as a musical instrument. Consequently, the primary functional obligations of each of these engineered artifacts have nothing to do with other potential functionalities. In contrast, a foliage leaf must perform multiple functions simultaneously, as for example light-interception, gas exchange, mechanical stability, defense against herbivory or pathogen attack, hydraulic requirements, heat dissipation, and a number of other developmental “obligations” attending its maturation from a small mass of primordial cells emerging from an apical meristem. In passing, however, it is useful to note that “unintended” functionalities (e.g., the use of a comb as a musical instrument or a hair drier to defrost a windowpane) provide evolutionary opportunities. Leaves probably evolved primarily as photosynthetic structures, but over the course of organic evolution they have become modified to yield spines, bracts, sepals, petals, stamens, carpels, and digestive organs.

In contrast, each of the engineering “standard practices” cannot be applied when dealing with biological structure–function relationships. Most organisms exist in a highly variable environment (e.g., air- or water-flow can vary many orders of magnitude, even in a few minutes), thereby requiring the biomechanicist to examine the “work-space” empirically. Perhaps worse, function has to be inferred. In some cases, this is not a huge challenge. The functionality of a snake’s hypodermic-like fangs seemingly requires little interpretation. As noted in the introduction, the behavior of a snake’s fang is sufficient to infer its function, arguably even in the absence of understanding it structure. Indeed, one can argue that “structure” cannot be used to infer “function” because “structure” implements function, whereas behavior is “function”. This is a sagacious dictum emerging from the “no function in structure principle” that reappears in various guises in both engineering theory and practice. This principle argues that function resides in the implementation of function, i.e., function resides in the behavior of a system. However, in many cases, biological “behavior” may be entirely absent. For example, what is the function of a rose’s prickle (the structure that is incorrectly called a “thorn” on a rose stem is actually a prickle and not a thorn)? Many believe that it functions as an herbivore deterrent. Yet, it can also function to dissipate heat, or as a grappling hook (indeed, prickles seem to be most developed in cultivars of climbing roses). Arguably, the rose prickle need not have any biological function whatsoever to provide a model for a better grappling hook or thermal vent. In this sense, a prickle can serve as a “model” for designing a device that has nothing to do with its biological functionality. However, such a model provides no mechanistic insights into the evolution and biology of a rose’s “thorn”. Consequently, in the context of this review, such a model is not a model—it is a prototype or pattern-architype.

Other problems present themselves to the biomechanicist. Structural configurations and tissue (material) compositions are constrained by evolutionary and developmental features that cannot be changed easily. As mentioned earlier, many biological structures also have multiple functions. Thus, although an engineer can try to maximize the performance of a machine performing one function, the biologist must interpret structures that most likely evolved to reconcile contrasting functional obligations.

It cannot escape attention that there is also a rather important contrast between physics and biophysics (Table 1). In most instances, one accurate measurement of a fundamental process or object (e.g., electrical resistance or the mass of an electron) with reliable equipment is sufficient to predict the behavior of the same process or object universally. A photon of a particular wavelength will behave the same way universality. In contrast, the accurate and precise measurement of the weight of a gerbil only reveals the weight of a very particular animal on a very particular day.

The foregoing may seem overly simplified and somewhat abstract. Certainly, all the limitations outlined in Table 1 can be overcome by the biomechanicist with sufficient insight and effort. That is not the real issue. What is important is dealing with the assumptions emerging from applying engineering theory to a biological problem. In order to illustrate this concern, we turn to a very simple case study—the application of engineering pertaining to the construction of a vertical support member to understanding the functionality of a tree trunk (Table 2).

With very few exceptions, the equations used to calculate the maximum load a columnar support member assume that the member has a uniform, unflawed geometry and that it is composed of a more or less uniform material. These equations also assume that the support member is subjected to a uniform loading condition (or that it experiences a principal stress application) under uniform working conditions. Most of these assumptions are implicit in the mathematics underlying the classic Euler–Greenhill formula used by many to estimate maximum tree height:$$H_{crit}=C{\left(\frac{E}{\rho g}\right)}^{1/3}D^{2/3},$$
where *H_crit_* is the theoretical critical buckling height of a column, *C* is a normalization constant that describes the uniform taper of the column (assumptions 1 and 2; see Table 2), *E* is Young’s modulus (assumption 3), *ρ* is the density of the construction material (assumption 3), *g* is the acceleration of gravity (assumptions 4 and 5), and *D* is the basal diameter of the column (assumptions 5 and 6). Curiously, the Euler–Greenhill formula makes another (often neglected) assumption—it assumes that the maximum height of a column is limited for (static) mechanical reasons rather than some other limiting factor. In the case of a tree, the maximum height is often limited by the rate at which water can be elevated to the upper reaches of the canopy at a rate sufficient to sustain stomatal conductance (and thus sustain the growth of first-year shoots).

## 6. Technological Challenges in Biomimetic Engineering

From the engineering perspective, once provided with underlying principles and understanding established in the modeling of organic systems, a further challenge emerges: to reproduce their functionality given available technologies. Although we live during a time in which technology has dramatically altered the human condition, our “age of technology” is infinitesimal in the context of evolution’s “engineering” of organic systems. Human-engineered materials and processes are, correspondingly, often quite crude and primitive when compared with organic systems with similar functionality. In many cases, even when we understand and can suitably model the structure and function of organic systems, the complexity inherent in their operation is beyond the capability of currently available technology to emulate [3].

As an example, efforts to recreate the biologically well-understood “muscular hydrostatic” structure of cephalopod arms and similar organs [26] in robotic variants in the early 2000s (e.g., [27,28]) were significantly limited by limitations in the functionality of artificial muscles. Mammalian muscles, which form the core structure of cephalopod arms (as well as elephant trunks and human tongues, among other familiar structures) are capable of strains well beyond that of their artificial equivalents. When combined with inherent limitations in the scalability, strength, and power generation of artificial muscle materials, the robotic versions of cephalopod arms produced lacked the complexity and hence several of the key movements (for example significant extension combined with torsion) that make their natural counterparts so effective (Figure 1).

Similar limitations have confronted engineers attempting to synthesize robotic versions of plants which can mimic the growth of structures such as roots and stems, in order to explore and examine imprecisely known environments. Some approaches have deployed thin telescoping tube structures [14]. However, “growing” such structures requires movement, relative to the environment, of large portions of the structure—the translating tube(s)—which in turn creates significant frictional resistance along the backbone when in contact with the environment. Plants avoid such issues by growing at the tips of their structures. Thus, environmental friction arising from plant growth is reduced to that at the tip. Experiments to mimic and exploit this behavior by depositing (3D printing) material directly at the tip of “robot roots” have been conducted [30]. However, the size of the technology currently required to 3D print suitable materials drastically limits the scalability of the resulting structures, with diameters at or below the size of the human finger not feasible at the present time.

It is worth noting that even when the structure/function relationships of organic systems are well understood, directly mimicking the organic structure may not meet the design goals of the engineer. Consider the oft-quoted argument against biomimetics, that if aviation engineers had restricted their efforts to reproducing the flight of birds, we would not today be flying at high speeds in airplanes. In the development of fixed-wing aircraft, the underlying principles of powered flight (notably the aerofoil wing profile), rather than the organic structures demonstrating it, proved the key to successful engineered systems. However, there are many situations in which not only the function but also the form of organic structures needs to be replicated via engineering (e.g., in prosthetics), and others in which either the underlying principles are poorly understood, and for which biomimetic engineering can be used to explore those principles. In this work, we focus on such situations, and in the following discuss in more detail inherent difficulties arising from mismatches in assumptions in standard practices of engineering, biomechanics, and physics in contrast to the environmental and evolutionary constraints arising in biophysics.

## 7. Criteria and Challenges for Assessing a Model

As noted, the use of modeling is generally unavoidable in almost every scientific and analytical discipline. Consequently, it is necessary to evaluate their performance empirically, which requires explicit metrics with which to measure the concurrence of a model’s predictions and the empirical data. In many cases, the metrics are de minimis statistical (e.g., 95% confidence levels) (Table 3). Under some circumstances, other metrics are necessitated (e.g., measuring mechanical or robotic performance and similitude to a biological prototype). In this context, it is essential to specify the scope of a model, thereby drawing a sharp distinction between prediction and extrapolation. By its very nature, the scientific enterprise typically attempts to predict the behavior of a system beyond the scope of the data used to model a system (Table 3). However, attempts to extrapolate a behavior or to draw conclusions beyond the extent of empirical observations or existing statistical trends are confronted with the danger that the methodological approaches or the range of behavior or trends will not continue beyond the scope of a model, i.e., interpolation within the scope of a model, differs profoundly from extrapolation beyond the scope of a model. Yet, another important test of a model is whether properties or behaviors emerge from the model that are not explicitly or directly engrained within the model’s structure. These criteria and challenges to validating a model surface across almost all biomechanical and biomimetic studies because most models are based on three generic assumptions: (1) the assumption that a single process is responsible for the pattern or behavior observed (either in the data or a physical manifestation of the system to be modeled), (2) the related assumption that one major process dominates an observed pattern or behavior, and (3) the assumption that biotic interactions are correctly interpreted and easily dissected. We have already noted that in some cases even an inaccurate model can be useful when it can be used by engineers or designers to create a useful result. For example, the model used in [31] to analyze observed circumnutations in plants, while not directly reflecting plant growth, has been used to effectively model the extension capability of spring-loaded vine-inspired robots [14]. However, as noted previously, the word “model” in this case refers to a “prototype” and differs substantively from a “model” intended to reveal mechanistic insights into the entity being modeled.

Setting aside the scientific challenges that face modeling a biological system, in our combined experiences there exists a sociological challenge. It seems clear to us that, in any non-trivial situation for which a biomimetic or biomechanical approach is motivated, it is essential to actively engage and combine the expertise of engineers and biologists drawn from the relevant specialties of each domain. One obstacle to progress in such intended synergies can be “culture shock” across domains. Engineers, when first confronted with the details of organic systems, tend to find the complexity of such systems quite intimidating. They can find the challenge of interpreting the available knowledge to extract underlying principles relevant to their engineering options overwhelming and rather demotivating. Experts in biology can be disappointed that the engineering options are often primitive and limited compared to organic structure and processes. Often, particularly in early prototypes, the engineered analogues offer somewhat sterile abstractions of the organic systems with which biologists are familiar. Enthusiasm and continued engagement requires open-mindedness, persistence, and good communication between all parties, which can sometimes be a challenge for people conditioned to specialized ways of thinking.

## Figures and Tables

**Figure 1 biomimetics-06-00021-f001:**
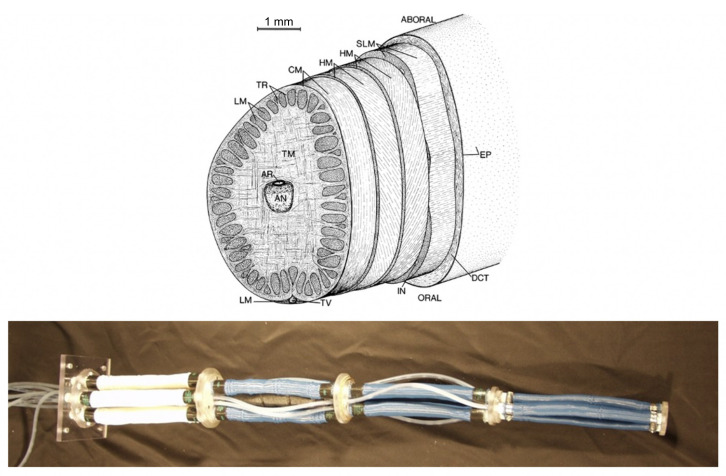
(**Top**): cross section of cephalopod arm musculature [29]; (**bottom**): robotic “octopus arm” [27].

**Table 1 biomimetics-06-00021-t001:** Comparisons between the standard practices of engineering vs. biomechanics.

Engineering	Biomechanics
1. Working environment specified a priori	1. The environment is variable
2. Design specifications are known	2. The environment has to be examined
3. Function is specified a priori	3. Function is inferred ad hoc
4. Structure and materials can be altered	4. Structure and materials have historical legacies
5. The structure typically has one function (thus, the function can be maximized)	5. The structure typically has multiple functions (thus, functions are reconciled)
6. Often, only a comparatively few accurate measurements are required to determine structural or material properties *	6. Many measurements are required because of natural biological variations in structure or materials

* The accurate measurement of the mass of an electron can be used to estimate the mass of all electrons in the universe, whereas the accurate measurement of the mass of a gerbil provides the mass of a single animal at a particular time in its life.

**Table 2 biomimetics-06-00021-t002:** Six assumptions (and qualifications) made when engineering a columnar support member.

∘ Assumption 1: Specified Geometry (no Growth Variations)
Qualification: *This is typically, but not invariably true. “Growth” (e.g., extension via controlled sliding of—themselves non-growing—elements) is almost always prespecified, and usually part of the design specifications for desired functionality.*
∘ Assumption 2: No production flaws (no fractures, or “knots”)
Qualification: *None. This assumption is almost universally made.*
∘ Assumption 3: Uniform (more or less) material composition
Qualification: *This assumption is almost always made, though more recent work in composite materials necessitates a relaxation of this assumption.*
∘ Assumption 4: Uniform stress application (compression, tension, or torsion)
Qualification: *Typically, one stress application (the extreme case) dictates the design, e.g., gravity or some uniform field applied to the whole member(s), or a small, finite number of applied point forces/moments, such as a point contact with the environment during drilling.*
∘ Assumption 5: Below yield stress conditions (designed not to break)
Qualification: *This assumption is typically tested* ad hoc *in physical prototypes* via *destructive testing.*
∘ Assumption 6: Uniform workplace conditions (controlled climate)
Qualification: *This is true for most engineering applications, but with some spectacular exceptions. For example, in Space systems in orbit, temperatures change over ranges of many hundreds of degrees in fairly short periods of time, and those systems have to be designed with those constraints in mind.*

**Table 3 biomimetics-06-00021-t003:** Three criteria for evaluating models and the challenges they present.

Criteria	Challenges
1. Fit simulation to empirical data (e.g., correlation and cross-validation analyses)	1. Specify metrics to measure concurrence (e.g., specify a—values and *r*^2^—values)
2. Specify the model’s scope (e.g., interpolation analyses)	2. Draw a sharp distinction between prediction and extrapolation
3. Identify emergent properties	3. Avoid the fallacy of causation

## Data Availability

Not applicable.

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
