# Peer review of "The Challenges of Inferring Organic Function from Structure and Its Emulation in Biomechanics and Biomimetics"

_biomimetics, 2021, doi:10.3390/biomimetics6010021_

Round 1

Reviewer 1 Report

The papers intends to be a philosophical statement describing and characterizing high-level challenges to using plant organisms as analogs for developing engineering solutions. I am struggling to find a contribution beyond what has been described by others as the general challenges of biomimetics. There are a few observations about plant life being different than animal life, however the challenges discussed apply to both. The challenges related to analog transfer and understanding are the same. For example, the challenges are domain perspectives, difference in the purpose and scope/applicability of models between biology and engineering, and the technology limits of trying to use electro-mechanical solutions for what are largely bio-chemical natural solutions. Ultimately, the paper does not provide and “proofs and conjectures” that would be clearly helpful in biomimetic application. I did find it a useful and enjoyable summary of the challenges that have been pointed out previously.

The following are some specific issues with regard to the perspectives discussed in this paper.

The first problem I have is the perspective that function must be inferred from the structure in plants and animals so that function can be achieved in biomimetic engineered solutions. This is not consistent with most engineering design theory that has some variation of the “no function in structure” principle. That is, from an engineering design perspective, functions are achievable by behaviors (the changing of parameters or properties) and behavior is implemented by structures. The challenge identified here is a result of skipping an acknowledgement of the behavior step. Using the snake fang analogy, the authors use the phrase “hypodermic-like”. This is the missing behavior description. The fang has very little structural resemblance to a medical hypodermic needle. However, the pressure and fluid flow behaviors are similar. It would be very difficult to infer function from structure without that behavioral description. The rose example is solved by calling them climbing support prickle or a heat dissipation prickle. It is thus the behavior that is key to finding useful analogs in organisms not inferring function from structure.

The second issue, is the point that engineered products largely have one function compared to organisms. This is a matter of perspective or abstraction. At a high level, the toaster’s function is to make toast. But also, at a high level, the objective of every organism is survival to successful reproduction. In this case, the opposite argument (from the author’s) about optimization can be made. The engineered product has a multi objective optimization problem where the organism only needs to maximize probability of successful reproduction. The toaster is not designed to maximize the amount of toast made or maximize the level of toastedness. Instead, the toaster has multiple objectives, even at the highest level of abstraction.

Finally, regarding the use of engineering models. The example of the buckling equation for the analysis of trees is not applicable (as the authors note). But the problem isn’t with the equation and the assumptions but rather using the wrong tool for the job. There are buckling modeling approaches for cellular or composite structures. These are not as elegant as Euler’s formulation, but they would require less of the challenged assumptions.

I agree with the author’s descriptions of model validation criteria and challenges but I would argue there should be more metrics for a model with respect to the application of biomimetics. The criteria listed relate to the accuracy of the model but not necessarily the model’s usefulness. For example, if a model inspired an engineer or designer to generate a novel (and ultimately successful solution), even if the model is inaccurate, I would argue it still has value and was useful.

Author Response

Dear Reviewer:

My co-author and I appreciate your comments and concerns about our manuscript, particularly your struggle with finding something new in what we have to say. Although others have addressed the challenges and opportunities that biomimetics presents, we felt that our comments about the limitations of modelling (in general) and mimicking biological systems (in particular) were worthy of consideration.

Clearly, you do not agree and we feel that nothing more that we can say would convince you otherwise. For this reason, my co-author and I have agreed to withdraw our submission and not seek to publish it elsewhere. If we cannot convince a reviewer such as you, we believe it would be hopeless to pursue our work further since others will likely share your judgment.

Thank you for your insights.

Cordially yours,

Karl Niklas (and Ian Walker)

Reviewer 2 Report

This is a nice commentary and I only have a few concerns. 

(1) The use of the word "optimized" in the abstract: I can say this as an engineer who got yelled at by a biologist for using the same word in the same type of (biomimetics) context.  Said biologist pointed out to me that nothing in nature is "optimized" because there is only an evolutionary driving force to change until something adequately fulfills the requirement.  This can include spectacular failure in some instances when overall the survival of a species is on average improved.  Thus there is no reason to say anything in biology is optimized in any way and the use of this word should be very careful in this context. 

(2) Table 1 is confusing and misleading.  It is unclear why there are so many comments made for engineering/biomechanics but only one gross generalization for physics/biophysics. But regardless, it just does not hold up to scrutiny.  When I saw the comment for Physics "One accurate measurement is sufficient" I immediately thought of the recent efforts that were made to experimentally "find" the Higgs Boson, which really is not consistent with the authors' point.  Perhaps one can make an accurate measurement of something well-established in historical terms (for example, Newtonian Mechanics) but we would not have universities full of scientists still studying physics if it was "done" science as implied by this epic simplification. 

(3) The discussion of the example/case study would really benefit from a simple diagram to accompany the text.  

(4) Consistent with my comments above about the authors tendency to oversimplify, I take issue with the statement on page 7 that a list of items serve only one function.  Who has not used a comb as a musical instrument?  I use my hairdryer for craft projects more often than for drying hair.  The paper would benefit dramatically from the authors backing down on strident assertions and leave them free to make their points in a more shades of gray instead of binary framework. 

Author Response

Dear Reviewer:

My co-author and I appreciated your comments and suggestions to improve our contribution to the plant biomimetics special issue. We were particularly intrigued by your biology teacher criticizing the use of the word ‘optimization’, particularly since one of us (KJN) has worked on computer simulations showing that much of early land plant evolution can be predicted (or at least is consistent with) the optimization of conflicting biological design specifications (e.g., maximizing the orientation of leaves for photosynthesis requires reconciling the maximization of bending moments).

My co-author and I believe that we could address each of your comments positively and constructively. However, we have decided to withdraw our submission because the second reviewer felt that we had nothing new to say. This criticism was revealing to us (it indicates that others will feel the same). Rather than publish an ‘inferior’ engagement in the pros and cons of modelling (and the challenges biomimetics presents), we felt it is prudent to withdraw with some semblance of intellectual dignity.

Thank you for your thoughtful review. I appreciate the time it took to prepare.

Cordially yours,

Karl Niklas (and Ian Walker)

Reviewer 3 Report

The manuscript "The Challenges of Inferring Organic Function from Structure and Its Emulation in Biomechanics and Biomimetics" by Karl Niklas and Ian Walker deals with a subject that is of high interest for everybody who is modelling biological structures and functions especially if aiming for a transfer to bioinspired materials and structures. The manuscript is well written, well understandable and discusses all relevant aspects, and I have nothing of a conceptual nature to criticize.

The authors succeed in putting many aspects and questions that constantly arise in the field of modelling complex biological structures and functions (shown exemplarily for biomechanics) and their transfer into biomimetic products into a larger context, pointing out possible pitfalls in the modelling and transfer processes, and suggesting solutions for avoiding these errors. For this reason, I am convinced that many researchers working in these fields will read the paper with great interest and profit. The tables give a concise summary of the points raised and discussed by the authors in the text, and will be very helpful for everybody working in this science areas but especially for scientists who want to start research in the interdisciplinary field of biomimetics.

The manuscript "The Challenges of Inferring Organic Function from Structure and Its Emulation in Biomechanics and Biomimetics" by Karl Niklas and Ian Walker deals with a subject that is of high interest for everybody who is modelling biological structures and functions especially if aiming for a transfer to bioinspired materials and structures. The manuscript is well written, well understandable and discusses all relevant aspects, and I have nothing of a conceptual nature to criticize.

The authors succeed in putting many aspects and questions that constantly arise in the field of modelling complex biological structures and functions (shown exemplarily for biomechanics) and their transfer into biomimetic products into a larger context, pointing out possible pitfalls in the modelling and transfer processes, and suggesting solutions for avoiding these errors. For this reason, I am convinced that many researchers working in these fields will read the paper with great interest and profit. The tables give a concise summary of the points raised and discussed by the authors in the text, and will be very helpful for everybody working in this science areas but especially for scientists who want to start research in the interdisciplinary field of biomimetics.

There is only one minor think I would ask the authors to take into consideration: Some of their well selected explanatory statements and examples, as e.g. the potential multifunctionality of the rose prickle or the height limitation of trees due to water transport, would win through appropriate quotations. I am aware that his might add some more citations to the manuscript, but I would like the authors to consider this aspect.

Based on the arguments raised above I suggest acceptance of the manuscript in its present form (perhaps with some more quotations).